# A Seroepidemiological Survey of *Corynebacterium pseudotuberculosis* Infection in South Tyrol, Italy

**DOI:** 10.3390/pathogens11111314

**Published:** 2022-11-09

**Authors:** Astrid Bettini, Marzia Mancin, Matteo Mazzucato, Angelica Schanung, Stefano Colorio, Alexander Tavella

**Affiliations:** 1Istituto Zooprofilattico Sperimentale delle Venezie, 35020 Legnaro, Italy; 2Scuola Superiore Provinciale di Sanità “Claudiana”, 39100 Bolzano, Italy

**Keywords:** *Corynebacterium pseudotuberculosis*, caseous lymphadenitis, caprine breeding, GEE models

## Abstract

*Corynebacterium pseudotuberculosis* is the causative agent of caseous lymphadenitis, a bacterial infection that can affect livestock. This infection can cause low growth rates and milk yields and reproductive failure, along with the infection of humans, especially those in close contact with the animals. The aim of this study was to evaluate the local CLA prevalence, highlighting the parameters for the potential predisposition to infection in goats, and to implement a monitoring program based on the newly acquired scientific evidence. Of a total of 2365 goat farms in South Tyrol, 384 farms were selected for the present study. A statistically significant number of animals were subjected to serologic analysis for the detection of *C. pseutotubercolosis* antibodies. A total of 2948 goats belonging to the selected farms were tested for CLA, 713 of which showed a positive result. The data analysis led to an estimated CLA seroprevalence of 21.85%. The results achieved can enable the evaluation and possible implementation of a voluntary program that permits us to test a larger number of animals using serological techniques. This program would be of great importance, especially for those farms dedicated to the production of milk and dairy products, as some manufacturing practices may increase the risk of transmission of zoonotic pathogens such as *C. pseudotuberculosis* to humans.

## 1. Introduction

Caseous lymphadenitis (CLA) is one of the most prevalent bacterial infections in livestock. It is an infectious chronic disease caused by *Corynebacterium pseudotuberculosis*, a facultative intracellular bacterium [1]. The main clinical signs of infection are soft caseous lesions in one or more superficial lymph nodes. The affected lymph nodes are primarily the superficial lymph nodes in goats and the lymph nodes within the chest and the internal organs, including the lungs and spleen, in sheep. The abscesses can later fistulate and discharge infectious necrotic contents into the environment, leading to possible transmission through fomites [2]. Other potential and more frequent routes of transmission are the oral mucosa and skin wounds. The disease can occur in two main forms, the cutaneous and visceral, which may coexist within the same animal [3]. Animals affected by CLA experience low growth rates and milk yields, as well as reproductive failure, while the infection of whole flocks can lead to great economic losses [2]. CLA is frequently observed in sheep and goats, but the occurrence of sporadic infections has also been described in New and Old World camelids [4], antelopes [5], buffaloes [6] and cervids [7]. Furthermore, several studies have highlighted the possible transmission to humans, especially those in regular contact with livestock [2,8,9].

The Autonomous Province of Bolzano in South Tyrol (Italy) is the northernmost Province of Italy and is geographically and morphologically marked by a pronounced subalpine to alpine terrain. At the time of the study design, the local goat population numbered approximately 26,368 individuals held in 2365 farms (data from the 2018 Provincial Database). All the goats older than 6 months of age are routinely sampled as part of the mandatory CAEV eradication program and processed for the detection of anti-SRLV antibodies through serological analyses [10]. The local government has also established a voluntary program against CLA, designed to be carried out within the CAEV eradication program. The current diagnostic technique is the palpation of the superficial lymph nodes [11].

The aim of this study was the assessment of the local CLA seroprevalence through the use of a serological analysis in order to evaluate the implementation of a surveillance program based on the newly acquired scientific evidence. Furthermore, an in-depth analysis of the breed, gender and age of the animals was conducted in order to evaluate their potential predisposition to infection based on the selected parameters.

## 2. Results

A total of 2948 goats belonging to 377 randomly selected farms were tested for CLA (Figure 1). A total of 186 out of the 377 farms (49.34%) and 713 out of the 2948 animals (29.18%) were positive for *C. pseudotuberculosis* antibodies (Figure 2 and Table 1). The number of goats per farm ranged between 1 and 181, but the majority (70%) of farms presented less than 10 animals. The percentage of positive goats per farm ranged between 5 and 100%, and 191 out of the 377 farms (50.66%) did not present CLA-positive goats.

The empty model (a model with only intercepts) indicates that the estimated CLA prevalence is 21.85% CI_95_ [19.32; 24.62]. The type III test applied to the multivariable model indicates that the variables of age and breed and the interaction between them are significant (Appendix A), whereas sex and the interaction between sex and the covariates are not significant. The AUC value is 0.7051 (Appendix A).

The significance of the interaction between the variables of breed and age indicates that the risk of being positive for CLA changes with age in a different way depending on the breed. As the animal’s age increases, the risk of being positive for CLA increases as well, depending on which breed is considered (Appendix A). At the same time, a unit increase in age increases the CLA prevalence differently according to which breed is selected.

The interpretation of the odds ratio suggests that, for any farm, the odds of an animal being positive for CLA among Chamois Colored goats is 2.2 times greater than that of crossbreed goats at the average age of 4.2 years. With the age increasing by one unit (5.2), the odds increase to 2.56. For a Passirian goat, the odds of being CLA-positive is 1.49 times greater than that of crossbreed goats at the average age of 4.2 years. With the age increasing by one unit (5.2), the odds increase to 1.82. Moreover, for a one-unit increase in age, increases of 22% and 28% in the odds of being positive for CLA can be expected in Chamois Colored goats and in Passirian goats, respectively (Figure 3).

When analyzing the results in terms of prevalence, the Chamois breed has the highest CLA prevalence: 29.68% CI_95_ [20; 41.61] at 4.2 years of age. This value increases to 34% CI_95_ [22.79; 47.21] with the age increasing by one unit. On the other hand, the breed with the lowest calculated prevalence is the crossbreed, with a value of 16% that does not increase with age (Figure 4).

## 3. Discussion

Caseous lymphadenitis (CLA) is one of the most relevant bacterial infections in livestock that can lead to low growth rates and milk yields, as well as reproductive failure, while the infection of whole flocks may have a great impact on economic losses [2]. While this infection is known to primarily affect cattle, small ruminants such as goats and sheep can become infected as well. The Autonomous Province of Bolzano in South Tyrol (Italy) implemented a voluntary program for the detection of lymphadenitis caused by *C. pseudotuberculosis* in goats in 2007. The aim of the voluntary program is the detection of clinical cases of CLA through the palpation of superficial/subcutaneous lymph nodes [11]. As this is only a voluntary program, which allows for the detection of clinical manifestations and not the infection status of the animal, data on the prevalence of *C. pseudotuberculosis* infections were available neither at the beginning nor after several years of the implementation of the program. For this reason, the authors decided to design a study for the collection of such data in order to evaluate the implementation of a more specific monitoring program through the use of a serologic diagnostic tool for the detection of CLA infections. Along with determining the CLA prevalence, another aim was to conduct an in-depth analysis of the potential predisposition to infection by studying several parameters, such as the breed, gender and age of the animals selected for the study.

During the study, 2948 goats belonging to 377 randomly selected farms were tested for the detection of anti *C. pseudotuberculosis* antibodies, 24.18% (49.34% at the farm level) of which tested positive. When analyzing the distribution of positive farms in South Tyrol (Figure 2), we found that the majority of them are located in areas with a high density of farms, particularly larger farms (Figure 1). Several factors may contribute to the positive rate in this particular location. These areas are characterized by private trading practices, without necessarily taking into consideration the sanitary health status of the animals, which may lead to the trade of subclinical infected animals. For the breeding of goats, the same buck is often used by several farmers, meaning that in the presence of an unknown infected buck, the infection can be spread to goats belonging to other farms. Furthermore, these areas are characterized by the absence of individual alpine pastures; therefore, goats from different farms are grazed together during the summertime, allowing for close contact between goats with an unknown CLA sanitary status. Many of the large seropositive farms are farms that are not devoted to milk production. These farms allow their animals to graze on alpine pastures during the summer months, while goats devoted to milk production remain on their farms all year long. This aspect could also be added to the other factors contributing to the presence of seropositive farms in the specific geographic area of South Tyrol. Finally, the typical local practice of transhumance, where the animals are moved from their original farms to the community alpine pasture, where they may share the same feeding and drinking locations, may also play a significant role in the spreading of CLA infections.

The statistical analysis conducted during this study indicates a CLA prevalence of 21.85% among the tested animals. Furthermore, the Chamoius Colored goat is the species with the highest prevalence (29.68%), while crossbreed goats have the lowest prevalence (16%). This analysis also highlights that, when the age variable increases by one unit, the prevalence values increase as well, indicating that an older population would most likely present with a higher prevalence than the population tested in the present study. Even though this serological assay (ELISA) is known to have some limitations with regard to the true prevalence, several studies have highlighted its strengths when comparing it to other diagnostic tools [12,13]. True prevalence values might not be achieved with this laboratory tool, even though the literature states that this serological analysis is sufficiently accurate under field conditions for positive CLA diagnosis in serology-based culling programs [13,14]. When evaluating other studies in the literature, this particular ELISA was compared to another ELISA and immunoblot method [15]. The ELISA used in this study did not perform as well as the other evaluated laboratory tools. This particular aspect might have impacted the prevalence values achieved in this study with regard to the true prevalence.

The results achieved and the data collected during this research project were presented to the local authorities in order to evaluate and possibly implement a voluntary program based on the newly acquired scientific evidence that would allow for the testing of a larger number of animals using serological techniques, instead of relying on the palpation practice. This program would be of great importance, especially for those farms dedicated to the production of milk and dairy products. Raw milk, without pasteurization practices, is often used for the production of dairy products, increasing the risk of transmission of zoonotic pathogens such as *C. pseudotuberculosis* to humans.

## 4. Materials and Methods

### 4.1. Study and Sampling Design: No Data on CLA Prevalence in South Tyrol Was Available

For this reason, the worst-case scenario of 50% CLA positive cases at the farm level was selected in order to calculate the number of farms for the sampling. Taking into account a level of confidence of 95%, an accuracy of 5% and a total of 2365 goat farms in South Tyrol, 384 farms were selected for the present study. The existing farms were organized into 5 groups based on the number of animals present on each farm (G1: 1–20; G2: 11–40; G3: 41–50; G4: 51–100; G5: >100). For each class, a sample of farms was selected by applying the simple random sampling method, without the replacement method. The number of farms to be tested was defined by taking into account the number of farms available within each group. Not all the farms selected for the present study were actually sampled due to technical difficulties at the time of sampling. All the data are presented in Table 2. The number of goats to be tested on each farm was determined in order to obtain a 95% certainty of the inclusion of at least one positive animal, given a hypothetical intra-farm prevalence of 20% [16]. Veterinarians were told to select the appropriate number of goats to sample on each farm, taking into consideration both males and females of a variety of ages. Function SURVEYSELECT in the SAS 9.4 software was used for the random selection and to provide the list of farms to be tested.

### 4.2. Serological Analysis

Blood serum samples were collected as part of the CAEV eradication program. The blood samples were drawn from the jugular vein using vacuum blood collection tubes with a clotting activator manufactured by Vacutest^®^ KIMA (Azergrande, Italy) and conferred to the Laboratory for Serology and Technical Assistance of the Institute for Animal Disease Control (Istituto Zooprofilattico Sperimentale delle Venezie—IZSVe) in Bolzano. Upon acceptance, all the blood samples were centrifuged at 3500 rpm for 3 min to obtain the serum. The serum samples were tested with a commercially available Enzyme Linked Immunosorbent Assay (ELISA) by IDScreen^®^ CLA Indirect (ID.vet Innovative Diagnostics, Grables, France). The sensitivity and specificity of the test according to the manufacturer are 100%.

### 4.3. Statistical Analyses

As animals may be either positive or negative for CLA, the status of the animal is a dichotomous outcome variable. A binomial probability distribution for the responsive variable was therefore assumed, and the model results were provided as both odds ratio and the prevalence. Since the data were obtained from goats clustered on farms, the outcomes were correlated [17,18]. Two types of model are appropriate for this type of data: the subject specific model using random effects (also referred to as the generalized linear mixed methods model, GLMM) and the population-averaged model using generalized estimating equations (GEE). Both models allow the researcher to estimate the CLA prevalence and to provide both the odds ratio as a measure of the associations between the covariates (sex and breed as the categorical variables and age as the continuous variable) and the probability of a goat being positive for CLA. Nevertheless, the subject-specific model (GLMM) enables the estimation of the risk of a particular farm being positive for CLA, whereas the population-averaged model (GEE) allows for an estimation of the risk of any farm being positive for CLA [2,19,20,21]. According to this explanation, the GEE model seemed to be more appropriate for the aim of this study, which was the evaluation of the CLA prevalence in order to implement a more appropriate surveillance program. Therefore, an empty GEE model (a model with only intercepts) and a GEE multivariable model including age, gender (F, M), breed (Chamois Colored goat, Passirian goat, Saanen, crossbreed) and their interactions were considered. In order to evaluate the significance of the factors specified in the model, the type III-F test was considered. The area under the ROC curve (AUC) was used to assess the goodness of the proposed model. AUC assumes values between 0 and 1, and higher AUC values are preferred [22]. A *p*-value of <0.05 was considered as statistically significant. The GENMOD functions in the SAS 9.4 software were used to fit the population-averaged models. SGPLOT was used to produce the graphical outputs [23,24].

## Figures and Tables

**Figure 1 pathogens-11-01314-f001:**
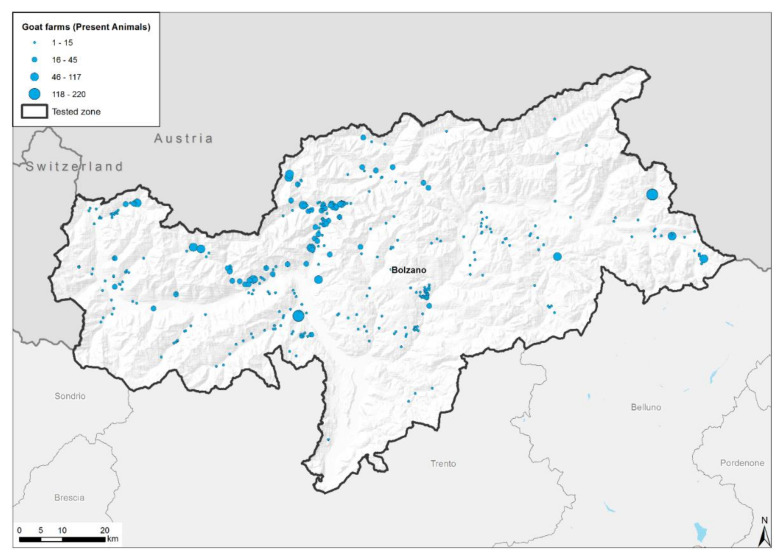
Distribution of tested farms in South Tyrol.

**Figure 2 pathogens-11-01314-f002:**
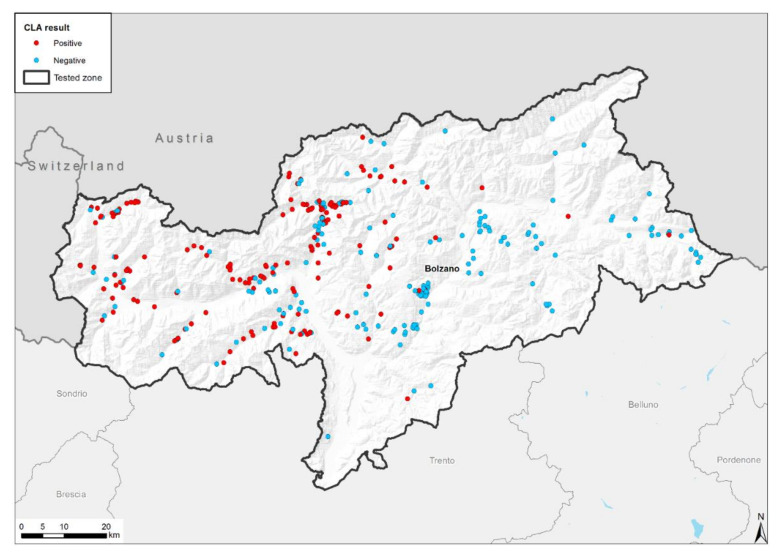
Distribution of CLA-positive (red) and -negative (blue) farms in South Tyrol.

**Figure 3 pathogens-11-01314-f003:**
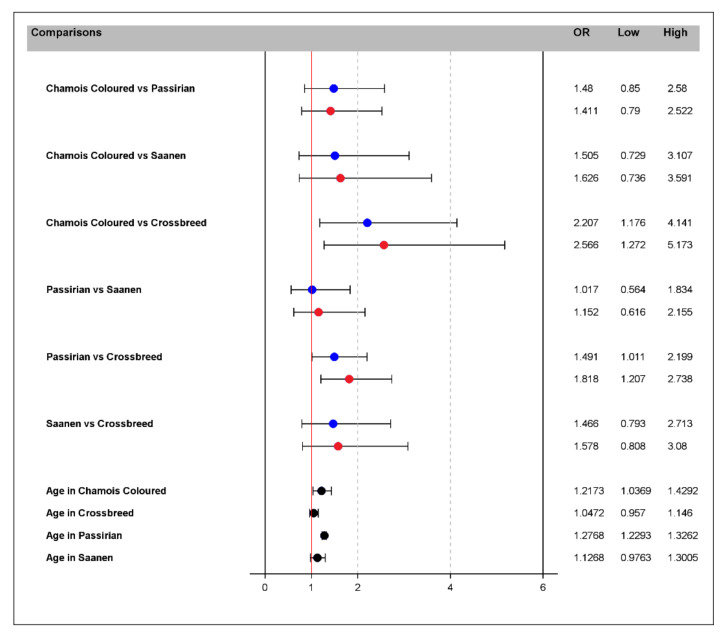
Odds ratio (OR) and 95% confidence intervals (low-high) based on the multivariable GEE model for the relationships between the covariates (breed, age and their interaction) and presence of CLA. Blue points indicate the odds ratio between breeds at the average age of 4.2 years. Red points indicate the odds ratio between breeds with the increase of 1 year in age (5.2). Black points indicate the odds ratio of a unit increase in age per breed. The red line shows an odds ratio = 1, indicating no difference in the risk of having CLA.

**Figure 4 pathogens-11-01314-f004:**
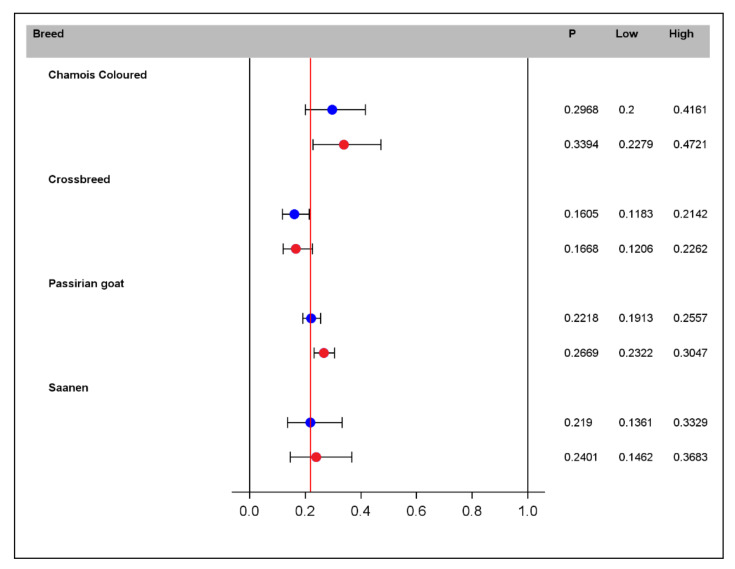
Estimated prevalence of CLA (P) and 95% confidence intervals (low-high) by breed and age based on the multivariable GEE model. Blue points indicate the prevalence at the average age of 4.2 years. Red points indicate the prevalence with the increase of 1 year in age (5.2). The red line shows the overall prevalence of CLA estimated using the empty model.

**Table 1 pathogens-11-01314-t001:** Number of CLA-positive and -negative animals per breed.

Breed	Number of Negative Goats	Number of Positive Goats	Total Number of Goats Tested
Passirian goat	1621	521	2142
Crossbreed	355	75	430
Chamois Colored goat	143	91	234
Saanen	109	25	134
Other	7	1	8
**Total**	**2235**	**713**	**2948**

**Table 2 pathogens-11-01314-t002:** Distribution of the number of farms selected during the study design and number of farms actually sampled.

Number of Goats/Farm	Total Number of Farms	Number of Selected Farms	Number of Sampled Farms
G1: 1–20	2004	325	318
G2: 21–40	248	40	40
G3: 41–50	35	6	6
G4: 51–100	59	10	10
G5: >100	19	3	3
**Total**	**2365**	**384**	**377**

## Data Availability

Not applicable.

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
