# Peer review of "A Seroepidemiological Survey of Corynebacterium pseudotuberculosis Infection in South Tyrol, Italy"

_pathogens, 2022, doi:10.3390/pathogens11111314_

Round 1

Reviewer 1 Report

"Subclinical diagnosis of Caseous Lymphadenitis caused by Corynebacterium pseudotuberculosis in South Tyrol – Italy" by Bettini et al is reviewed.  The manuscript provides a seroprevalence survey of goats in a region of Italy.  This type of data is important, as there is a limited understanding of the prevalence of CL in many regions and there may be important associations with breeds or management practices that may be elucidated.  Strengths: The paper has a robust sample set from diverse populations of animals.  Weaknesses: The methods and results are a bit muddled or incomplete.  Was there any testing for associations between farm size or geographic region?  Are there any management practices that could be evaluated?  I think the paper would be stronger if additional variables besides age, sex, and breed were considered, since some of this like farm size appears to have been collected. The discussion should also include discussion of the limitations of the serological assay, for example what is known about this assay performance that might affect true prevalence?  

Specific Comments: 

Line 58- It is unclear in both methods and results how these farms/samples were randomized.  Please describe how this was done.

Line 69 - what is empty model?

Line 74-86 this seems like a lot of discussion instead of results.  Please state the results before interpreting them.

Table 1 and 2- can you include the n for the sex and species as well as farm size and breed?

Line 176- 178- Please indicate how this test was used and interpreted.  From my understanding CL serological testing is notorious for having low sensitivity and overall test performance when compared with culture and other results.  Please indicate this as a limitation or how this potential was assessed.  

Author Response

  • No statistical analysis was carried out in regard to the association between farm size and geographic region. A GIS map was provided to show where the sampled farms were located in South Tyrol. Furthermore, Figure 1 shows the location and size of farms, whereas Figure 2 shows the location of the seropositive farms in our Province.
  • Management practices were not evaluated in the present study. The only aspect the authors can relate to is that the majority of large farms is located in the Passiria Valley; these farms are extensive farms not devoted to milk production. Animals devoted to milk production are not allowed to go on alpine pastures during the summer months, while non lactating goats are. On alpine pastures, goats from different farms are in close contact for several months, increasing the risk of transmission of any pathogen between animals and therefore farms as well. This particular aspect could explain why the majority of seropositive farms are present in geographical areas where animals are mainly bred for hobby reasons and not milk production. This information was added to the manuscript as well (lines 138-143).
  • Unfortunately, no other variables were taken into account when designing this study. Data on farm size is available, but not strong enough to provide further statistical analyses.
  • A brief discussion on the limitations of the serological assays was provided (lines 152-158).

Specific comments:

  • Line 58: the sampling method was further explained in the Materials and Methods section. The  farms were organized into 5 groups, based on the number of animals present within each farm (G1: 1-20; G2: 11-40; G3: 41-50; G4: 51-100; G5: >100) and for each class a random sample of farms was selected applying the simple random sampling without replacement. Each farm is identify by a code (000BZ000).  A progressive number from 1 to  n (n=maximum number of farms for each class) was linked to each farm code  to simplified the selection of farms by SAS software. In detail, the SAS function "SURVEYSELECT" was applied to select the desiderate number of farms (defined proportionally according to the number of farms available for each class) in a random way, without replacement, providing the list of farms to test for CLA, identified by farm code. As far as the selection of individual goats within the farm is concerned, veterinarians were told to select the appropriate number of goats to sample within each farm taking into consideration both males and females of a variety of ages. Function SURVEYSELECT in SAS9.4 software was used for random selection to provide the list of farms to be tested.
  • Line 69: the empty model is a model with only intercepts. It is empty because it doesn’t have any explanatory variables. This specification was added in both the Materials and Methods section and Results section of the paper.
  • Lines 74-86: this brief description was added in order to allow the reader to understand the next couple of sentences of results. If the reviewer prefers for this to be moved to another section it can be done.
  • Tables 1 and 2: tables were modified accordingly
  • Lines 176-178: a discussion on limitations of this technique was added, supported by literature added both in text and in the Reference section of the manuscript.

We thank the reviewer for all revisions.

Reviewer 2 Report

The Authors describe the results of a seroepidemiological survey of Corynebacterium pseudotuberculosis infection in goats from a well defined geographical area.

The paper needs to be improved. Here below, specific comments that can help in the revision process are reported

General comments

- The title includes the terms “Subclinical Caseous Lymphadenitis”. Since serological data are reported only, it cannot be excluded that some animals had a clinical form (eg, enlargement of subcutaneous lymphnodes). Moreover seropositivity indicate an immune response following the infection, not necessairly associated to a disease. In conclusion I suggest to modify the title in “Seroepidemiogical survey of Corynebacterium pseudotuberculosis infection ……………………”.

- No comparison is made between serological data and data coming from the palpation of the superficial lymphnodes: since the voluntary program against CLA started in 2019 some data should be available. This should be disussed (see below).

- Main features the serological test (specificity, sensitivity) are not given (see below).

Specific comments

- Line 31 – after “signs” there is a colon (:) instead if a point (.): this error is repeated in other parts of the manuscript, please check carefully.

- Lines 31-33 – The sentence is incorrect. Rephrase it.

- Lines 106-113 – This is a repetition of what is reported in the introduction.

- Line 115 – C. pseudotuberculosis > in italics

- Lines 139 -142 – This is a repetition of what is reported in the results section. In the Discussion part the results should be explained, interpreted, compared to those of other studies, …………..

- Line 154 – “No data on CLA prevalence in South Tyrol was available.” This sentence should be moved in the line below.

Moreover: a. data from the voluntary program against CLA started in 2019 are not available? b. There are no data from nearby geographic areas

- Lines 176 -178 – More details regarding the serological test are needed.

Author Response

General comments:

  • The title was modified according to the reviewer suggestion.
  • Unfortunately, no data concerning the voluntary program, including palpation results, was provided. The study design of the present work was supposed to include a comparison with those results, along with other laboratory analyses. Unfortunately due to lack of both personnel and resources, this was not possible. The authors apologize for the misunderstanding of this concept.
  • Features of the serological test were added to the manuscript.

Specific comments

  • All colons were modified to point. We apologize for the errors
  • Lines 31-33: The sentence was rephrased
  • Lines 106-113: The authors are aware of the repetition, it was rephrased in order to provide some background to the discussion. If the reviewers believe this is not necessary, these sentences can be taken out of the manuscript.
  • Line 115: C. pseudotubercolosis was modified to italics
  • Lines 139-142: the sentece referring to the results was written in order to be able to discuss it without creating any confusion on what the authors were talking about. If needed, it can be eliminated from the manuscript. Unfortunately, there are not a lot of other studies presenting CLA seroprevalence results, and a comparison to other studies or other regions was not possible.
  • Line 154: the sentence was moved to the appropriate paragraph. We apologize for the typing mistake.
    • Data from the voluntary program was never really made available. The program actually started in 2007 and veterinarians used the palpation method to evaluate the potential presence of abscesses in those farms that voluntarily took part to the program. In the presence of a clinical form (presence of abscesses), the animal was euthanized and the farm was presented with a monetary compensation. Since the beginning of the plan, the clinical form is more or less absent in South Tyrol. Unfortunately, no actual data was ever produced from this program and is therefore unavailable for comparison.
    • To the authors’ knowledge, there are no similar plans carried out in the nearby Italian regions. Unfortunately, comparison of such data was not available either.
  • Lines 176-178: data on sensitivity and sensibility were added.

We kindly thank the reviewer for all revisions.

Round 2

Reviewer 1 Report

The author's have mostly addressed all the comments.  My only minor comment is line 197 that the performance of this test is 100% sensitivity and specificity.  Is this clinical specificity and sensitivity?  If so this seems suspect. Please clarify what these values are and where they are from.  Please see DOI: 10.2376/0005-9366-16024 where this assay was evaluated in animals in the clinical context.  

Author Response

The presented sensitivity and specificity values were provided directly from the manufacturer’s website. A clarification sentence was added in the discussion section (lines 158-161), taking into consideration the literature provided by the reviewer, which was added as citation both in text and in the reference section.

We kindly thank the reviewer for his/her work. It was very much appreciated

Reviewer 2 Report

The manuscript is now acceptable for publication.

Author Response

We kindly thank the reviewer for his/her work. It was very much appreciated